# A Fully Remote Diagnostic and Treatment Pathway in Patients with Obstructive Sleep Apnoea during the COVID-19 Pandemic: A Single Centre Experience

**DOI:** 10.3390/jcm10194310

**Published:** 2021-09-22

**Authors:** Andras Bikov, Safia Khalil, Meg Gibbons, Andrew Bentley, David Jones, Saba Bokhari

**Affiliations:** 1Regional Sleep Service, Wythenshawe Hospital, Manchester University NHS Foundation Trust, Manchester M23 9LT, UK; safiakhan10@yahoo.it (S.K.); megsduncan@hotmail.co.uk (M.G.); andrewbentley@nhs.net (A.B.); david.jones5@mft.nhs.uk (D.J.); saba.bokhari1@mft.nhs.uk (S.B.); 2Division of Infection, Immunity & Respiratory Medicine, University of Manchester, Manchester M23 9LT, UK

**Keywords:** obstructive sleep apnoea, continuous positive airway pressure, COVID-19

## Abstract

The COVID-19 pandemic led to significant delays in the diagnostic and management pathway of patients with obstructive sleep apnoea (OSA). During the first wave of COVID-19, our department adopted a fully remote diagnostic (home cardiorespiratory polygraphy) and treatment (autoset continuous positive airway pressure, CPAP) approach. As a novel mode of service delivery, our aim was to evaluate our pathway and analyse factors associated with adherence to CPAP. We analysed the first 300 patients (51 ± 13 years, 48% men) who were set up on remote CPAP between 20 May 2020 and 11 September 2020. The associations between CPAP usage at 90 days and age, gender, body mass index, disease severity, Epworth Sleepiness Scale and comorbidities were investigated with linear and logistic regression analyses. A total of 124 patients (41.3%) were fully-adherent to CPAP therapy, defined as CPAP usage ≥ 4 h on ≥ 70% of the days. Only driving status was associated with adherence to CPAP. Patients who were adherent at 28 days were more likely to stay adherent at 90 days (3.77 odd ratio /3.10–4.45/ 95% confidence interval). We have shown that a fully remote diagnostic and treatment pathway for patients with OSA can be successfully delivered, and our preliminary outcomes of adherence to CPAP are comparable with published data.

## 1. Introduction

Obstructive sleep apnoea (OSA) is a common disease which is characterised by repetitive complete or partial collapse of the upper airways during sleep. Emerging evidence suggest that untreated OSA is a risk factor for COVID-19, including severe disease and death [1]. Most importantly, this risk might be mitigated by long-term continuous positive airway pressure treatment [2]. Therefore, timely diagnosis and treatment are of clinical importance. However, the number of diagnostic tests has been reduced by 80% in Europe during the first wave due to staff relocation, staff sickness and strict disease prevention policies [3].

Recognising the risk of transmission of COVID-19, recommendations from various professional societies and experts have encouraged home-based testing for OSA [4,5,6,7]. Of note, home cardiorespiratory polygraphy has been recommended by the American Academy of Sleep Medicine as a diagnostic sleep test for patients with high pre-test OSA probability and low-risk (i.e., controlled cardiovascular and respiratory disease) even before COVID-19 [8] and the National institute for Health and Care Excellence (NICE) guidelines [9] recommend it as a first line test in diagnosis of OSA.

Continuous positive airway pressure (CPAP) is the first line treatment for OSA. However, it is an aerosol generating procedure [10], and therefore, inpatient CPAP setup needs special protective measures. Coupled with the need for hospital visits for diagnostics, setup and follow up, this can potentially increase the risk of transmission of COVID-19 to patients and staff. This risk can be reduced if the treatment is set up via an ambulatory pathway [7].

During the first wave of the COVID-19 pandemic, our department adopted a fully remote diagnostic and treatment approach for patients referred with suspected OSA. In this article, we describe our pathway, analyse factors associated with adherence and present the results of a patient survey.

## 2. Materials and Methods

### 2.1. Study Design and Subjects

We analysed the first 300 patients who were set up on remote CPAP between 20 May 2020 and 11 September 2020. Patients who declined treatment or those who were set on non-invasive ventilation were not included in this analysis. The standard operating procedure for our remote diagnostic and CPAP pathway for OSA is summarised in Figure 1.

Patients were referred with symptoms suggestive for OSA, including snoring, pauses in breathing, excessive daytime sleepiness and comorbidities frequently associated with OSA such as obesity, cardiovascular disease or diabetes. Following triage by trained sleep physiologists, patients received cardiorespiratory polygraphy via home delivery through a courier service, and they were supplied with written instructions and an online video outlining set up of the equipment. They returned their device on the following day together with a completed questionnaire reporting their subjective sleep quality and quantity on the night of the analysis and an Epworth Sleepiness Scale (ESS). The test was scored by trained sleep physiologists based on the American Academy of Sleep Medicine criteria.

Following the sleep study, the patients were triaged into either a consultant or senior sleep physiologist led clinic. The outcome was based on the burden of comorbidities, symptoms suggestive for other sleep disorders on the referral or the presence of a significant sleep disordered breathing other than OSA (i.e., central sleep apnoea, hypoventilation syndrome) identified on the sleep study. These more complex patients were seen by the consultants.

Patients were contacted by telephone where during a 15 min consultation a detailed medical history was taken, and the diagnosis of OSA and the treatment were explained. In line with the NICE guidelines [9], patients with moderate-to-severe OSA and those patients with mild OSA and significant daytime symptoms were offered a CPAP device. An information package containing a template for a face mask was delivered to those patients who agreed to remote CPAP setup. An AirSense™ 10 AutoSet device (ResMed UK) device with the corresponding mask was then delivered via courier to the patient’s home with written instructions and an online video outlining the setup process. Ongoing telephone support was delivered by sleep physiologists. The usage of CPAP, effective pressures and residual apnoea-hypopnoea index (AHI) was monitored with the Airview software (ResMed UK) at 2, 7, 28 and 90 days after initiation, and patients were contacted by a trained sleep physiologist if the compliance was not adequate. The patients were encouraged to contact our helpline during working hours. The humidification and ramp period were defined based on patients’ feedback at 2 days. Auto-set was the default modality with 4–20 cmH_2_O minimum and maximum pressure as default settings. The pressure was fixed at 95% of maximal pressure for those patients who complained about residual sleepiness or had a significantly high AHI. All patients had full-face mask (the size was based on the template) and humidifier (level 4) at baseline. The mask type and the level of humification were adjusted if necessary, during the follow up. An anonymised patient survey was sent 21 days following treatment initiation. The survey collated feedback on difficulties experienced by patients, sources of help they looked for and changes in symptoms following remote CPAP treatment.

As this was a service evaluation project, the Institutional Research Office exempted the approval from the Research Ethics Committee.

### 2.2. Sleep Test

The cardiorespiratory polygraphy was performed with the Nox T3 Portable Sleep Monitor (ResMed, Didcot, UK) device which measured nasal airflow, pulse oximetry, thoracic and abdominal effort, snoring and body position. Apnoea was defined as at least 90% reduction in the nasal flow lasting for at least 10 s. Hypopnoea was defined as at least 30% reduction in the nasal flow associated with at least 3% drop in oxygen saturation [11]. An AHI ≥ 5/h was diagnostic for OSA; mild OSA was defined by an AHI 5–14.9/h, moderate by an AHI 15–29.9/h and severe by an AHI ≥ 30/h.

### 2.3. Statistical Analysis

The JASP 0.14 (JASP Team, University of Amsterdam, Amsterdam, The Netherlands) software was used for statistical analysis. The normality of the data was checked with the Shapiro–Wilk test. The adherent and non-adherent groups were compared with *t*-test, Mann–Whitney and Chi-square tests. Factors associated with adherence at 90 days were analysed with linear regression and *t*-test (when adherence was defined as the average hours of usage) and bivariate logistic regression (when a patient was considered adherent to therapy if they used their device ≥4 h on ≥70% of the days) [12]. Recognising that continuous variables may not necessarily be related linearly to adherence, patients were divided into different groups based on age (<31, 31–50, 51–70 and >70 years), excessive daytime sleepiness (ESS < 11 and ≥ 11), BMI (<25, 25–30, 30–35 and >35 kg/m^2^) and AHI (5–14.9, 15–29.9, ≥30 events/hour). The prevalence of adherence was compared with Chi-square test among subgroups. Factors associated with change in adherence between 28 and 90 days were studied with logistic regression analysis. We compared characteristics of patients seen by senior physiologists or consultants with *t*-test, Mann–Whitney test and Chi-square test. Data are presented as percentages, mean ± standard deviation, median /interquartile range/ or odds ratio (OR) /95% confidence interval/. A *p* < 0.05 was considered significant.

As this was a service evaluation project, no formal a priori power analysis was performed. Post hoc sensitivity analysis revealed that analysing 300 patients allowed us to identify ≥ 1.66 OR for factors associated with adherence at 90 days with a power of 0.80 and α error of probability of 0.05 [13].

## 3. Results

### 3.1. Comparison of the Adherent and Non-Adherent Groups

The CPAP usage at 2, 7, 28 and 90 days is summarised in Table 1.

Thirteen patients (4.3%) had never switched their machine on, and 124 patients (41.3%) were fully-adherent to CPAP therapy at 90 days. There was no difference in age, BMI, gender, the prevalence of comorbidities, the severity of OSA, ESS between the adherent and non-adherent groups (all *p* > 0.05, Table 2), whilst the prevalence of drivers was significantly higher in the adherent group (*p* = 0.03). There was no difference in adherence depending on whether the patients were seen by a consultant or a senior physiologist.

### 3.2. Factors Associated with Adherence to Treatment at 90 Days

Only the absence of diabetes and driving status were associated with adherence to treatment at 90 days when assessed with bivariate logistic regression (Table 3). When both factors were analysed together using multivariate logistic regression, only the driving status continued to be significant (1.98 /1.04–3.77/, *p* = 0.04).

The average usage at 90 days was 3.9 ± 2.7 h. Neither age, BMI, AHI or ESS was related to this. There was no difference in the average usage when patients were stratified based on gender, comorbidities or even driving status (all *p* > 0.05).

There was no statistically significant difference in the percentage of adherent patients when the groups were stratified based on age (Figure 2A), excessive daytime sleepiness (Figure 2B) or AHI (Figure 2C, all *p* > 0.05). However, significant differences were observed in the prevalence of adherent patients based on BMI (Figure 2D, *p* = 0.04).

### 3.3. Factors Associated with Changes in Adherence between 28 and 90 Days

A total of 127 patients (42.3%) were adherent to CPAP at 28 days with an average usage of 4.0 ± 2.7 h. Eighteen patients became adherent between 28 and 90 days. Compared to those who remained non-adherent, there was no difference in any of the investigated parameters (all *p* > 0.05). In contrast, 21 patients became non-adherent between 28 and 90 days. Compared to those who remained adherent, there was also no difference in any of the investigated parameters (all *p* > 0.05). The odds ratio/95% confidence interval/for patients to be adherent at 90 days if they were adherent at 28 days was 3.77 /3.10–4.45/ *p* < 0.01).

### 3.4. Comparison of Patients Seen by Consultants or Senior Sleep Physiologists

There was no difference between the patients seen by consultants or senior sleep physiologists in the prevalence of males (*p* = 0.31), patients with diabetes (*p* = 0.81), gastro-oesophageal reflux disease (GORD, *p* = 0.16), chronic heart failure (*p* = 0.33), hypertension (*p* = 0.24), ischaemic heart disease (*p* = 0.11), cerebrovascular disease (*p* = 0.62), or depression/anxiety (*p* = 0.19). In contrast, consultants saw more patients with chronic airway disease (24% vs. 14%, *p* = 0.03) and atrial fibrillation (9% vs. 3%, *p* = 0.03), and there were more patients in the physiologist group who did not have any comorbidities (22% vs. 8%, *p* < 0.01). The percentage of drivers was higher in the physiologist group (84% vs. 71%, *p* < 0.01). BMI was higher in the consultant group (36 /31–43/ vs. 34 /30–39/, *p* = 0.02). There was no difference in age (*p* = 0.87), ESS (*p* = 0.89), AHI (*p* = 0.94) and most importantly average hours of usage at 90 days (*p* = 0.93) or the percentage of adherent patients (*p* = 0.74).

### 3.5. Results from the Patient Questionnaire

The patient survey was completed by 108 patients. Due to its anonymised nature, we could not connect these data with adherence to CPAP. The most common difficulties with using the treatment included mask discomfort, noise and adapting to a regular routine. The most common source for help included the MyAir application, internet search and direct contact to our service. More than 60% of patients felt less sleepy following the CPAP therapy, and 40% had fewer daytime naps. It was noted that 4% felt worse following the treatment, and 17% did not feel any benefit from the treatment. The responses are summarised in Figure 3.

## 4. Discussion

The COVID-19 pandemic has posed a significant challenge on sleep services. According to our knowledge, this is the first report describing a fully remote diagnostic, treatment and monitoring service for patients with OSA.

The overall adherence to CPAP therapy was 41%. This is comparable with levels of adherence for intention-to-treat analyses reported in large clinical trials [14,15]. However, this adherence is significantly less than reported in a large United States-based real-life study (75%) [16]. The difference could in part be due to anxieties regarding risk of transmission of COVID-19 to household contacts from community CPAP users [17] but also due to differences in reimbursement, as CPAP treatment is covered by the National Health Service in the United Kingdom [18]. Analysing the effect of the COVID-19 pandemic on CPAP usage, no change in adherence to CPAP has been reported in the United States [19,20]. In contrast, adherence to CPAP had increased in France [21] and Spain [22] during the first wave of the pandemic in Spring 2020. More concerning is that the incidence of insomnia in general population [23] and among patients with OSA [19] is increasing. Therefore, the adherence to CPAP therapy might decline on long-term. Although insomnia may compromise the adherence to CPAP among patients with OSA [24], we have not systematically evaluated insomnia symptoms in this database which is one of the limitations of the study.

CPAP is the most effective treatment for OSA. However, a variable level of adherence to CPAP treatment has been reported by various studies [18], and this tends to decline with long-term usage [25]. Numerous sociodemographic and disease related factors have been identified as predictors for long-term CPAP adherence [12], but the relationship between adherence and these factors was not always replicated by independent studies [18]. In our study, demographics, comorbidities, excessive sleepiness or disease severity were not related to adherence at 90 days. These results are similar to those found in the Sleep Apnoea cardioVascular Endpoint (SAVE) study that evaluated 12-month adherence to CPAP [25].

Disease severity and daytime sleepiness are the most extensively investigated factors in studies focusing on adherence to CPAP [12]. However, a systematic review concluded that these relationships are usually relatively weak and disappear following adjustment on confounding factors [12]. In our analysis, neither disease severity nor daytime sleepiness was related to adherence. However, it must be noted that in our pathway the severity of OSA was calculated based on a home cardiorespiratory polygraphy which underestimates AHI and is less precise than polysomnography. In addition, polysomnography could potentially reveal factors, such as sleep efficiency [26] or low arousal threshold [27] which could be useful to predict adherence to treatment. Nevertheless, changes in symptoms rather than baseline values relate to CPAP adherence [12]; however, we did not systematically record these in our analysis.

Mental health disease has frequently been investigated as a possible factor associated with adherence to CPAP treatment [12,28]. However, in line with the current literature [12], we did not find any relationship between mental health disease and adherence to CPAP. The presence of mental health disease was based on patient and referral reports, and it is possible they may have been underreported as has been the case with the incidence of cardiovascular disease which has been found to be significantly lower than has been reported in large multi-centre studies [29,30]. Of note, systematic assessment of anxiety based on questionnaires has not been performed in our study. A recent report by Celik et al. concluded that higher anxiety scores based on the Self-rating Anxiety Scale may predict non-adherence to CPAP [31] suggesting that the severity rather than the historical diagnosis of anxiety is associated with adherence. Although mental health diseases were not related to adherence, their assessment could still be vital when tailoring psychological intervention [17,28,32]. Higher BMI was previously found to be associated with better adherence to CPAP at 90 days [33]. Our results also highlight that the relationship between BMI and adherence is not linear. In general, the adherence to CPAP is similar in older people to other age groups [12,34]. In line with this, we did not find a significant difference in adherence among the various age groups. Our results conflict with a recent analysis of a large cohort from the United States which found age a significant predictor for CPAP adherence with patients aged 60–80 years being the most adherent (70–80% adherence rates) [35]. However, differences in reimbursement of CPAP between different countries needs to be considered when comparing the data. Nevertheless, our results are comparable to the PREDICT trial which showed that adherence to CPAP among patients over 65 years is 35% [36]. Of note, certain factors associated with older age, such as the presence of cognitive decline, insomnia, nocturia or benign prostate hypertrophy may be associated with lower adherence [12]; however, these were not investigated in the current study.

In contrast to other factors, driving status was the strongest predictor for treatment adherence. Driving has been recognised as an important factor associated with greater CPAP usage [37]. According to the governmental regulations in the United Kingdom, patients with OSA and excessive daytime sleepiness can continue driving only if they are treated [38]. However, fitness for driving is based on a comprehensive assessment which includes ESS, but not exclusively [39]. This could be a possible reason why driving rather than ESS was associated with better adherence. The Epworth Sleepiness Scale contains 8 questions exploring the propensity of falling asleep in different situations, including driving. Further studies, evaluating the impact of specific questions on adherence on CPAP are warranted.

In line with the results of the SAVE study [25], we found a strong correlation between the usage of CPAP at 28 days and 90 days. This highlights the importance of early identification of non-adherent patients and prompt attention to early interventions in this group. A personalised approach, education, behavioural interventions, telemonitoring and involving family and friends have been shown to improve adherence to CPAP [32]. The most common sources of help used by our patients were telehealth systems, phone calls to the service and the internet. Our patient survey identified numerous barriers with CPAP usage but most of these could be addressed with additional physiologist, psychologist or medical input. Approximately 20% of patients did not feel benefit or their symptoms became worse following CPAP therapy. There is a need for further diagnostics and consultations to focus on causes for persistent symptoms despite CPAP treatment.

It has been reported that the capacity of sleep laboratories decreased by 80% during the first wave of the COVID-19 pandemic [3]. As respiratory physicians were and continue to be redeployed to manage acutely ill patients, we maintained our clinical capacity by converting a significant amount of consultant clinics to senior physiologists-led clinics. We demonstrated that by careful selection of patients, a similar efficacy in treatment outcomes can be achieved in these clinics. We believe this example should encourage other services worldwide should they face a reduction in their capacity due to clinical circumstances.

## 5. Conclusions

We have shown that (1) a fully remote diagnostic and treatment pathway for patients with OSA can be successfully delivered and preliminary outcomes of adherence are comparable with published data. (2) The odds ratio of the patients who were adherent at 28 days was 3.77 (3.1–4.45) to remain adherent at 90 days. (3) The driving status was associated with adherence to treatment at 90 days. We anticipate further adaptations to the pathway driven by the COVID-19 pandemic will continue to improve our outcomes.

## Figures and Tables

**Figure 1 jcm-10-04310-f001:**
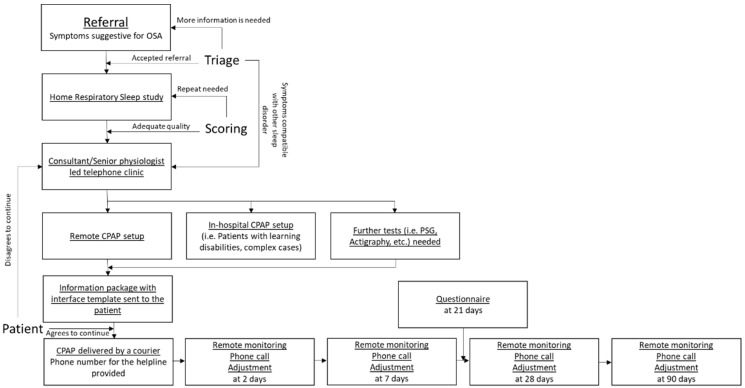
Standard operation protocol for the remote CPAP service. OSA—obstructive sleep apnoea, CPAP—continuous positive airway pressure.

**Figure 2 jcm-10-04310-f002:**
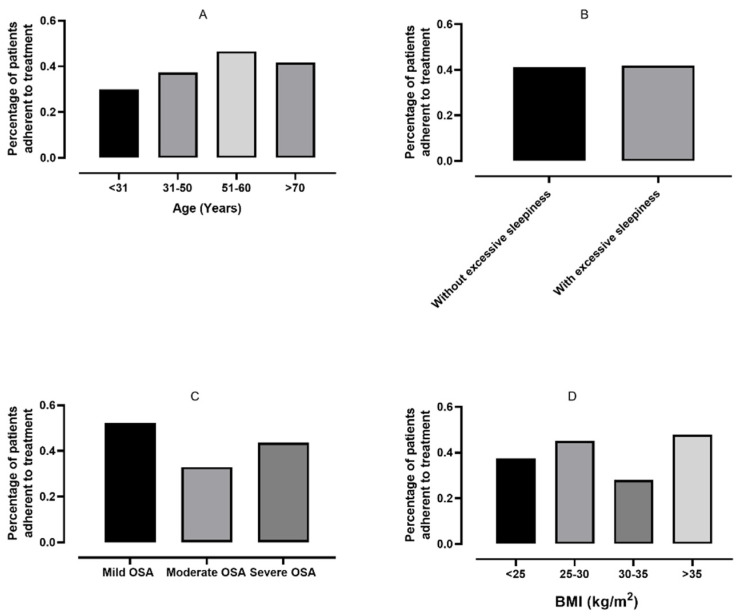
Comparison of adherence among different subgroups. Percentages of patients adherent to continuous positive airway pressure plotted based on age (**A**), excessive daytime sleepiness (**B**), OSA severity (**C**) and body mass index (BMI, **D**).

**Figure 3 jcm-10-04310-f003:**
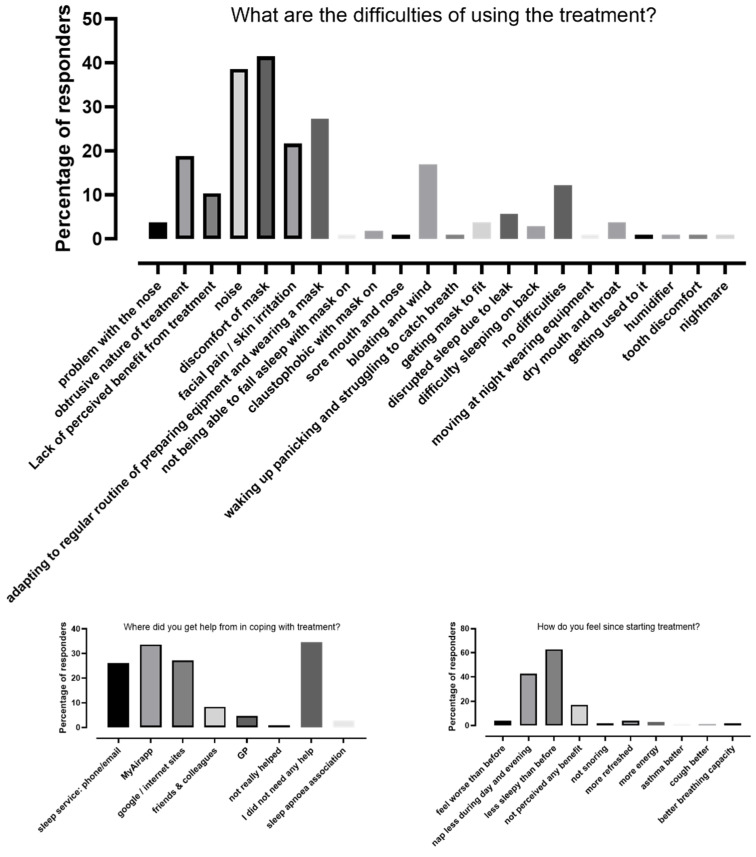
The result of the patient survey.

**Table 1 jcm-10-04310-t001:** Continuous positive airway pressure (CPAP) usage at 2, 7, 28 and 90 days.

	2 Days	7 Days	28 Days	90 Days
Used days Used days ≥4 h	2 /1–2/ 1 /0–2/	6 /3–7/ 4 /0–6/	25 /16–28/ 17 /4–24/	75 /42–87/ 52 /11.7–76/
Percentage of used days ≥4 h	50 /0–100/	57 /0–86/	61 /14–86/	58 /13–84/
Total hours used	7.3 /0.0–12.8/	29.0 /11.0–45.0/	121.2 /46.0–171.3/	373.5 /129.6–560.4/
Median usage (hours on days used)	4.4 /0.0–6.6/	5.0 /2.3–7.0/	5.4 /3.0–7.0/	5.5 /3.2–7.1/
Average usage (hours)	3.7 /0.0–6.4/	4.2 /1.4–6.4/	4.3 /1.6–6.1/	4.2 /1.4–6.2/

Data are presented as median /interquartile range.

**Table 2 jcm-10-04310-t002:** Comparison of the adherent and non-adherent groups.

	Total (*n* = 300)	Adherent (*n* = 124)	Non-Adherent (*n* = 176)	*p* Value
Age (years)	51 ± 13	52 ± 13	50 ± 14	0.11
BMI (kg/m^2^)	35 /30–40/	36 /30–40/	34 /30–40/	0.40
Gender (males%)	48	46	50	0.49
Chronic airway diseases (%)	18	17	18	0.78
Hypertension (%)	29	30	29	0.88
Ischaemic heart disease (%)	10	8	11	0.55
Cerebrovascular disease (%)	2	3	1	0.20
Atrial fibrillation (%)	6	7	5	0.45
Chronic heart failure (%)	4	4	4	0.76
Diabetes (%)	13	14	12	0.76
GORD (%)	13	14	12	0.76
Depression/anxiety (%)	16	17	15	0.37
No comorbidities (%)	16	15	16	0.78
ESS	12 /8–16/	12 /8–16/	12 /8–16/	0.95
AHI (events/hour)	35 /22–51/	39 /22–53/	33 /22–48/	0.17
Driver (%)	79	86	75	0.03
Seen by a consultant (%)	40	39	41	0.74

AHI—apnoea-hypopnoea index, BMI—body mass index, ESS—Epworth Sleepiness Scale, GORD—gastro-oesophageal reflux disease. Data are presented as mean ± standard deviation or median/interquartile range.

**Table 3 jcm-10-04310-t003:** Factors associated with compliance at 90 days.

	OR /95% CI/	*p* Value
Age (years)	0.98 /0.97–1.00/	0.12
BMI (kg/m^2^)	0.99 /0.97–1.02/	0.73
Gender (males)	0.85 /0.54–1.35/	0.49
Chronic airway diseases	0.92 /0.49–1.70/	0.78
Hypertension	1.04 /0.62–1.74/	0.88
Ischaemic heart disease	0.78 /0.35–1.76/	0.55
Cerebrovascular disease	2.95 /0.53–16.36/	0.22
Atrial fibrillation	1.47 /0.54–4.04/	0.45
Chronic heart failure	1.21 /0.36–4.06/	0.76
Diabetes	0.45 /0.22–0.92/	0.03
GORD	1.11 /0.55–2.24/	0.76
Depression/anxiety	0.75 /0.40–1.41/	0.38
No comorbidities	0.91 /0.48–1.74/	0.78
ESS	0.99 /0.95–1.04/	0.91
AHI (events/h)	0.99 /0.98–1.00/	0.25
Driver	2.01 /1.06–3.81/	0.03
Seen by a consultant	1.08 /0.67–1.76/	0.74

OR—odds ratio; OR with /95% CI/ are presented.

## Data Availability

Data are available upon direct request from the corresponding author.

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
