# Peer review of "A Fully Remote Diagnostic and Treatment Pathway in Patients with Obstructive Sleep Apnoea during the COVID-19 Pandemic: A Single Centre Experience"

_jcm, 2021, doi:10.3390/jcm10194310_

Round 1

Reviewer 1 Report

Dear Editor/Authors

This is a very interesting article about delivering a fully remote diagnostic and treatment pathway in OSA patients. The authors made a random selection of the correct population for their study, with inclusion and exclusion criteria leading to the success of the main target of the study. The statistical methodology was appropriate for the analysis of nominal and ordinal parameters. The results, described in a different section, provided clear and more informative data. Unfortunately, no main correlation succeeded.

The main conclusions of the study are the following:

  1. The diagnostic and treatment evaluation of OSAS will be successfully introduced in-home base statement.
  2. The odds ratio of the patients who were adherent at 28 days was 3.77 (3.1-4.45) to remain adherent at 90 days.
  3. The driving status was associated with adherence to treatment at 90 days.

Minor Comments

  1. It is important that all 3 aforementioned conclusions be introduced in the final conclusion of the study, giving publication potentiality to the study.
  2. The association between driving status and adherence was demonstrated. On the contrary, an association was not found between ESS and adherence. The authors made a comment on that – but did not correlate the last question of the ESS, namely “In a car, while stopped for a few minutes in traffic or at a light”, with adherence. It is of interest to identify this correlation as ESS is commonly used in all labs, and a special unique searching in that question may provide important information.
  3. Insomnia is a recognized symptom found in many patients with OSA. COMISA is the abbreviation describing the CO-Morbid of Insomnia with Sleep Apnea. The authors asked their patients about insomnia and made a comment in the Discussion Section – but they need to further correlate the existence of insomnia with adherence.

Author Response

Q1: The main conclusions of the study are the following:

  1. The diagnostic and treatment evaluation of OSAS will be successfully introduced in-home base statement.
  2. The odds ratio of the patients who were adherent at 28 days was 3.77 (3.1-4.45) to remain adherent at 90 days.
  3. The driving status was associated with adherence to treatment at 90 days.

 It is important that all 3 aforementioned conclusions be introduced in the final conclusion of the study, giving publication potentiality to the study.

A1: We thank the reviewer for their comments. We added the main conclusions in the revised manuscript. 

Q2: The association between driving status and adherence was demonstrated. On the contrary, an association was not found between ESS and adherence. The authors made a comment on that – but did not correlate the last question of the ESS, namely “In a car, while stopped for a few minutes in traffic or at a light”, with adherence. It is of interest to identify this correlation as ESS is commonly used in all labs, and a special unique searching in that question may provide important information.

A2: We fully agree that the individual questions of the Epworth Sleepiness Scale have different impact on both clinical burden and likely adherence to CPAP. Unfortunately, we do not have access to the raw questionnaires in many patients anymore, therefore we could not investigate this relationship in detail. We expanded the discussion with this relevant point.

Q3: Insomnia is a recognized symptom found in many patients with OSA. COMISA is the abbreviation describing the CO-Morbid of Insomnia with Sleep Apnea. The authors asked their patients about insomnia and made a comment in the Discussion Section – but they need to further correlate the existence of insomnia with adherence.

A3: Thank you for your comment. Co-existing insomnia can indeed compromise the adherence to CPAP in OSA (https://pubmed.ncbi.nlm.nih.gov/27976438/). However, insomnia has not been systematically evaluated in this cohort, therefore unfortunately, we could not analyse this. We expanded the discussion with this limitation.

Reviewer 2 Report

I commend the authors for this paper, documenting and providing outcomes from a telehealth service for the diagnosis and treatment of obstructive sleep apnea. The reported adherence rate (41%) seem to me to be on par with that reported with clinical services. The results from this study have potentially broad applicability to other sleep apnea clinics.

I do not have major issues with the paper, but have a few suggestions for details that could/should be added to the text

Would you provide more information (for example in a form similar to Figure 1 but modified to be a participant flow diagram) on the numbers referred, tested for sleep apnea (including proportion of tests that had to be repeated etc), diagnosed with OSA, recommended CPAP, needed in-hospital CPAP set-up as opposed to remote set-up? We know 300 patients were set up to commence CPAP remotely, but it would help to understand how applicable this pathway was to the patients attending the clinic as a whole.

Was there a protocol specifying criteria as to who should be recommended CPAP, based on their clinical features and the results of the respiratory polygraphy? This should be described.

How much time was spent interacting with the patients with education, setup and monitoring?

Table 2, please check the first column, I think the “(%)” should be omitted for the binary variables as no percentages are given. Coefficients from logistic regression are quoted under the column named Beta, but it would be more reader-friendly to have odds ratios (exp(beta)) and 95% confidence limits. I would do the same for the multivariate logistic regression findings were described in line 141.

Author Response

Q1: Would you provide more information (for example in a form similar to Figure 1 but modified to be a participant flow diagram) on the numbers referred, tested for sleep apnea (including proportion of tests that had to be repeated etc), diagnosed with OSA, recommended CPAP, needed in-hospital CPAP set-up as opposed to remote set-up? We know 300 patients were set up to commence CPAP remotely, but it would help to understand how applicable this pathway was to the patients attending the clinic as a whole.       

A1: Thank you very much for this important comment. We completely agree with the reviewer that these data are necessary to understand the whole service. However, the ratio of patients referred, treated, those who declined treatment and those who needed inpatient setup, NIV, etc. do not reflect on the current practice due to the following reasons: First, the referrals to our unit during the first wave have been significantly dropped (from 60 patients/week pre-COVID to 20 patients/week); this is currently 40/week. Second, patients experienced a significant delay for their first consultation due to non-essential services were completely closed during the first wave. Third, the capacity of in-hospital diagnostic and CPAP setup varied depending on the incidence of new cases. As the requested data do not reflect on the current practice and need to be interpreted with caution acknowledging governmental and local policies at that time, we decided not to include them. We currently evaluate these data as part of a quality improvement project.

Q2: Was there a protocol specifying criteria as to who should be recommended CPAP, based on their clinical features and the results of the respiratory polygraphy? This should be described.

A2: Patients with moderate-to-severe OSA, and those patients with mild OSA and significant daytime symptoms were offered a CPAP device. 2.1 section of the manuscript was expanded to contain this information. 

Q3: How much time was spent interacting with the patients with education, setup and monitoring?

A3: The post-polygraphy consultation in average takes 15 minutes. This was added to the manuscript. The detailed instructions on CPAP setup were sent via a post mail. For the monitoring we use the Airview software. Depending on the difficulties (i.e. mask, humidifier, claustrophobia, noise, etc.) the troubleshooting consultation may take 5-20 minutes. The latter figures were not systematically collected for this project.                                                                         

Q4: Table 2, please check the first column, I think the “(%)” should be omitted for the binary variables as no percentages are given. Coefficients from logistic regression are quoted under the column named Beta, but it would be more reader-friendly to have odds ratios (exp(beta)) and 95% confidence limits. I would do the same for the multivariate logistic regression findings were described in line 141. 

A4: Thank you. Table 2 and its figures were modified accordingly.